# Primary Aggressive Oral Lymphomas (PAOL): A Narrative Review of Diagnosis, Molecular Features, Therapeutic Approaches, and the Integrated Role of Dentists and Hematologists

**DOI:** 10.3390/cancers17193138

**Published:** 2025-09-26

**Authors:** Michele Bibas, Andrea Pilloni, Edmondo Maggio, Andrea Antinori, Valentina Mazzotta

**Affiliations:** 1Department of Clinical Research: Hematology, National Institute for Infectious Diseases “Lazzaro Spallanzani” I.R.C.S.S., 00149 Rome, Italy; 2Department of Oral and Maxillofacial Sciences, Sapienza University of Rome, Via Caserta 6, 00161 Rome, Italy; 3Department of Clinical Research: Infectious Diseases, National Institute for Infectious Diseases “Lazzaro Spallanzani” I.R.C.S.S., 00149 Rome, Italy

**Keywords:** primary aggressive oral lymphomas (PAOL), oral non-Hodgkin lymphoma, oral plasmablastic lymphoma (PBL), oral Burkitt lymphoma (BL), oral diffuse large b-cell lymphoma (DLBCL), oral peripheral T-cell lymphoma (PTCL), integrated diagnostic approach, viral co-infections (HIV; EBV; HHV-8), tumor microenvironment, genomics, dentist–hematologist collaboration

## Abstract

Primary aggressive oral lymphomas (PAOL) are a rare subset of non-Hodgkin lymphomas that occur in the oral cavity without any systemic involvement at diagnosis. They account for 2–3% of all lymphomas and are typically aggressive B-cell subtypes like large B-cell lymphoma and plasmablastic lymphoma. These malignancies often present with non-specific symptoms, leading to diagnostic delays. An integrated diagnostic approach combining oral examination, imaging, biopsy, immunohistochemistry, and genetic studies is crucial for accurate diagnosis and staging. Treatment usually involves systemic chemotherapy, rituximab for CD20+ tumors, and adjunctive radiotherapy for localized disease. Research into PAOL’s genomic and microenvironmental landscape is paving the way for targeted therapies. In HIV+ or transplant patients, PAOL may be driven by viral co-infections, requiring tailored therapy. Dentists play a crucial role in PAOL diagnosis, prevention, and treatment, and their collaboration with hematologists is essential.

## 1. Introduction

Primary aggressive oral lymphomas (PAOL) are very rare extra-nodal non-Hodgkin lymphomas that originate in the oral cavity without evidence of systemic disease at diagnosis [1]. They account for about 2–3% of all lymphomas, making them an uncommon occurrence in a location where ~90% of malignancies are squamous cell carcinomas [2]. Patients often present to dentists with nonspecific oral issues (gingival or palatal swelling, tooth mobility, or persistent ulcers) that mimic more common dental conditions [3]. Consequently, misdiagnosis or delayed diagnosis is common, which can adversely affect outcomes. Early detection and accurate diagnosis are therefore critical [4]. There are no specific treatment guidelines for primary oral lymphomas; management typically adheres to principles established for lymphomas in other locations [5,6]. We conducted a comprehensive literature search to identify studies on PAOL from 1990 through June 2025. The primary databases were MEDLINE/PubMed, with supplementary searches of Web of Science and Cochrane Library for any systematic reviews. Key search terms included combinations of “oral lymphoma,” “aggressive lymphoma,” “lymphoma,” “prognostic,” “survival,” and specific lymphoma subtypes (e.g., “DLBCL,” “BL,” “PBL,” “T-cell lymphoma”). This review provides a comprehensive overview of PAOL from a hematological standpoint, highlighting clinical-pathological characteristics, existing and novel therapies, and the crucial collaborative role of dentists in diagnosis, supportive care, and long-term follow-up.

## 2. Epidemiology

PAOL are rare in the general population, but their incidence varies by patient cohort and region. In immunocompetent individuals, primary oral NHL typically occurs in middle-aged to elderly patients (median age in the 60 s) with a slight male predominance (~1.3–1.4:1) [7,8,9,10]. In contrast, in areas with high HIV prevalence or among immunosuppressed patients, PAOL are more common and tend to present at younger ages [11]. Patients with HIV/AIDS have a markedly increased risk of NHL, and oral involvement is relatively frequent in this group [12,13]. Similarly, organ transplant recipients on chronic immunosuppressive therapy can develop EBV-driven oral lymphoproliferative disorders [14]. Overall, immunodeficiency (HIV infection or iatrogenic) is a significant contributor to oral lymphoma pathogenesis [11,12,13,14]. The distribution of histologic subtypes in PAOL mirrors that in aggressive lymphomas generally [15]. Diffuse large B-cell lymphoma (DLBCL) is consistently the most common subtype (~40–50% of cases), followed by plasmablastic lymphoma (PBL, ~20–30%). Burkitt lymphoma (BL) constitutes only a small percentage (~3–5%), usually in younger or immunocompromised patients [15]. Indolent B-cell lymphomas like extra-nodal marginal zone (MALT) lymphoma have been reported in the oral cavity, but these typically present differently and will not be the focus of this review. T-cell/NK-cell lymphomas are uncommon (≈5–10% of oral lymphomas); among these, peripheral T-cell lymphoma, NOS (PTCL-NOS) and extra-nodal NK/T-cell lymphoma (nasal type) have been documented, whereas anaplastic large cell lymphoma (ALCL) is extremely rare in the oral location [15,16,17,18,19].

Anatomically, primary oral lymphomas may involve soft tissues or bone (jaw). The gingiva and hard palate are the most frequently affected sites. In fact, the gingiva is the single most common site of PAOL, representing roughly one-third of cases. The palate (hard and soft) is another common site, followed by other locations like the alveolar bone of the jaws, tongue, buccal mucosa, floor of mouth, and tonsillar area. Soft tissue involvement is more common than bony involvement, but intraosseous jaw lesions do occur. Lymphomas in the jaw can manifest as poorly defined, “motheaten” osteolytic lesions on imaging, which may be misinterpreted as osteomyelitis or severe periodontal disease. Figure 1 illustrates common intraoral sites and frequencies of lymphoma subtypes, and Table 1 summarizes the anatomical distribution of the main PAOL subtypes.

## 3. Histopathologic and Immunophenotypic Features

Histologically, PAOL do not significantly differ from their counterparts in other sites, but recognizing the specific subtype via morphology and immunophenotype is crucial for proper classification and treatment [20,21]. The majority of PAOL are aggressive B-cell neoplasms. In case series, Diffuse large B-cell Lymphoma (DLBCL) is the most common histology (~40–50% of cases), followed by Plasmablastic Lymphoma (PBL) (~20–30%) [12,19,21]. Burkitt Lymphoma (BL), while very aggressive, comprises only a small percentage of oral cases (~3–5%), usually in younger or HIV-positive patients. T-cell lymphomas (TCL) represent about 5–10% of oral lymphomas (mostly PTCL-NOS or NK/T-cell lymphoma) [12,19,21]. Each subtype has characteristic microscopic morphology and an immunophenotype identifiable by immunohistochemistry (IHC) or flow cytometry [21]. (see Table 2 for key markers).

### 3.1. Diffuse Large B-Cell Lymphoma (DLBCL)

DLBCL is a diffuse proliferation of large, atypical lymphoid cells that efface the normal tissue architecture. In the oral cavity, it often presents as a submucosal mass or within bone marrow spaces of the jaw [20,21]. Histologically, sheets of large lymphoid cells with vesicular nuclei and prominent nucleoli are seen. Tumor cells express pan-B-cell markers (CD20, CD79a, PAX5) and typically lack plasma cell markers (e.g., CD138). The cell-of-origin can be germinal center B-cell (GCB) or activated B-cell (ABC) subtype, determined by expression of markers like CD10, BCL6, and MUM1/IRF4. The proliferation index (Ki-67) is usually high. Some oral DLBCLs, particularly in older or immunosuppressed patients, are EBV-positive (EBER in situ detectable in tumor cells). “Double-expressor” phenotype (co-expression of MYC and BCL2 proteins) may be observed and is associated with worse prognosis [20,21]. DLBCL is the most common PAOL subtype and essentially an oral presentation of the same disease seen in nodal sites.

### 3.2. Plasmablastic Lymphoma (PBL)

PBL is an aggressive B-cell lymphoma with plasma cell differentiation, strongly associated with HIV infection. It was first described in the oral cavity of AIDS patients. Morphologically, PBL shows diffuse sheets of large cells resembling immunoblasts or plasmablasts, often with eccentric nuclei, prominent nucleoli, and abundant basophilic cytoplasm. The immunophenotype is distinct: PBL cells typically lose B-cell markers (CD20, PAX5 are negative) while expressing plasma cell markers such as CD138, CD38, MUM1/IRF4, and PRDM1/Blimp-1 [20,21]. Clonal immunoglobulin light chain restriction may be demonstrable. CD79a may be positive despite CD20 negativity. Ki-67 is ~100%, indicating explosive growth. EBV is present in up to ~70% of PBL cases (EBER in situ positive). Because PBL can resemble a plasmacytoma or multiple myeloma, correlation with systemic findings and immunophenotypic differences help distinguish them [20,21].

### 3.3. Burkitt Lymphoma (BL)

BL is a highly aggressive B-cell lymphoma composed of a monomorphic population of medium-sized cells with round nuclei, multiple small nucleoli, and a very high mitotic rate. BL often shows a classic “starry sky” pattern with numerous tangible-body macrophages scattered among the tumor cells [20,21]. In endemic (African) BL of the jaw, the tumor cells show remarkable uniformity. BL tumor cells strongly express B-cell markers (e.g., CD20, surface IgM) and germinal center markers (CD10, BCL6), and nearly 100% of cells are Ki-67+. BL typically lacks BCL2 protein expression. Genetically, BL is defined by a translocation of the MYC gene on chromosome 8q24 to an immunoglobulin locus (usually IgH on 14q32); this t (8;14) translocation is present in ~100% of cases [20,21]. EBV is involved in most endemic BL (∼90%) but only a subset of sporadic BL (20–40%, intermediate in HIV-associated BL). In adults, an apparent isolated oral BL lesion is exceedingly rare—if one is diagnosed, extensive staging is needed because most sporadic adult BL cases have abdominal or marrow involvement.

### 3.4. T-Cell Lymphomas

Extranodal NK/T-cell lymphoma, nasal type, typically involves the mid-facial region (palate or nasal cavity) and can present with a destructive, ulcerative lesion that sometimes perforates into the nasal cavity. Histologically, it shows an angiocentric, necrotizing infiltrate of atypical lymphoid cells; the tumor cells usually express cytotoxic molecules (Granzyme B, TIA-1) and are nearly always EBV-positive by EBER in situ [20,21]. Peripheral T-cell lymphoma, NOS (PTCL-NOS) can rarely present in oral soft tissue or jaw; it is essentially a diagnosis of exclusion among mature T-cell neoplasms and often requires clonality studies for confirmation. Anaplastic large cell lymphoma (ALCL) is very uncommon as a primary oral lesion; ALCL consists of sheets of large, pleomorphic T-cells that strongly express CD30, and ALK-positive ALCL (seen in younger patients) will show ALK-1 protein expression [20,21].

## 4. Molecular Pathogenesis

Aggressive oral lymphomas exhibit several molecular abnormalities (Table 3) analogous to their nodal counterparts, encompassing distinctive chromosomal translocations and gene mutations that enable lymphomagenesis. Crucial genetic events that are critical are MYC translocation, BCL2 and BCL6 translocations, recurrent gene mutations, TP53 mutations, PRDM1 (BLIMP1) mutations, other B-cell lymphoma mutations, and IRF4 (MUM1) rearrangement.

MYC translocations are the defining characteristic of BL, resulting in unregulated MYC expression and promoting its highly proliferative phenotype. About 10% of DLBCL cases also have a change in the MYC gene [22]. Double-hit DLBCLs, which are “high-grade B-cell lymphomas with MYC and BCL2 and/or BCL6 rearrangements,” are more aggressive than typical DLBCLs and have substantially worse outcomes [20,21,22]. The identification of MYC and BCL2/BCL6 rearrangements with FISH is important for clinical practice considering double-hit lymphomas often require more aggressive chemotherapy than R-CHOP [22,23,24].

BCL2 and BCL6 translocations are prevalent in aggressive lymphomas, with BCL2 translocations present in a subset of cases and BCL6 translocations observed in approximately 20–30% [25]. These rearrangements may lead to treatment resistance or occur as part of double-hit lymphoma [23,24,25]. Aggressive lymphomas also have recurrent gene mutations, as well as mutations in the NF-κB pathway, TP53 mutations, and PRDM1 (BLIMP1) mutations [26,27,28,29]. Further a subset of MYC mutations is linked with negative outcomes [29]. B-cell lymphomas frequently exhibit other mutations in pathways regulating cell proliferation and survival, with activated B-cell DLBCL commonly harboring MyD88 and NF-κB pathway mutations [26,27,28,29].

The IRF4 (MUM1) rearrangement moves the IRF4 gene, which encodes the MUM1 protein. Large B-cell lymphoma with IRF4 rearrangement manifests in children or young adults within the head and neck regions, exhibiting an excellent prognosis despite high-grade characteristics [26,27,28,29]. Oncogenic viruses are important in many oral lymphomas, and Epstein–Barr virus (EBV) is involved in some types of PAOL [30]. Oncogenesis driven by EBV develops through latent viral proteins that promote proliferation and inhibit apoptosis [31]. Lymphomas associated with EBV are particularly prevalent in immunodeficient conditions and endemic Burkitt lymphoma [32,33]. Human Herpesvirus 8 (HHV-8) is responsible for Kaposi’s sarcoma and primary effusion lymphoma (PEL), but its involvement in PAOL is minimal [34]. Classic PBL is HHV-8 negative; however, infrequent oral plasmablastic neoplasms may exhibit HHV-8 positivity in HIV-positive individuals [35].

Chronic oral inflammation is an issue in the tumor microenvironment and oral microbiome that may help lymphomagenesis [36]. Epidemiological studies have identified a correlation between severe periodontal disease and an elevated risk of non-Hodgkin lymphoma (NHL). Oral lymphomas linked to EBV frequently exhibit elevated PD-L1 expression on both tumor and immune cell populations, potentially promoting local immune evasion [36,37].

## 5. Diagnostic Approach

Prompt and accurate diagnosis of PAOL requires an integrated, multidisciplinary approach. Typically, a dentist or oral surgeon identifies a suspicious lesion and performs a biopsy for histopathology, then a hematologist and radiologist carry out complete staging and management. Figure 2 outlines a diagnostic and initial treatment pathway.

### 5.1. Clinical Suspicion and Initial Evaluation

The clinical manifestations of PAOL are often nonspecific, contributing to frequent misdiagnosis. Nonetheless, certain warning signs in an oral lesion should raise suspicion for lymphoma. Common presentations include an unexplained localized swelling or mass (e.g., a rapidly enlarging gum swelling or a mass on the palate or tongue) that is often firm and sometimes painless. Ulceration of the mucosa overlying the mass is common as it enlarges, sometimes with necrosis and associated bleeding. Lymphoma invading the periodontium or jawbone can lead to loosening of teeth out of proportion to local periodontal status. Nerve involvement (e.g., inferior alveolar nerve) may cause paresthesia or a “numb chin” syndrome in mandibular lesions. Extra-oral extension can manifest as facial swelling or asymmetry corresponding to tumor spread or regional lymphadenopathy. Table 4 lists key red flags for PAOL (e.g., non-healing ulcer >2 weeks, unexplained tooth mobility, persistent localized pain or numbness) that help differentiate these from benign conditions.

### 5.2. Imaging of the Lesion

Imaging studies define the extent of the lesion but are often not diagnostic by themselves. Dental radiographs (panoramic X-rays) or cone-beam CT of jaw lesions may show ill-defined radiolucent areas of bone destruction; while suggestive of malignancy, these can be misinterpreted as osteomyelitis or advanced periodontitis. In soft tissue lesions, contrast-enhanced CT or MRI of the maxillofacial region typically demonstrates a mass that may infiltrate adjacent structures; unlike squamous carcinomas, lymphomas often produce more submucosal expansion with relatively intact surface mucosa until late ulceration. FDG-PET/CT is very useful for staging, as it will show avid uptake at the oral tumor and can reveal additional disease sites. Overall, imaging helps delineate the lesion’s extent and guides the biopsy to the optimal site.

### 5.3. Biopsy and Pathologic Confirmation

A definitive diagnosis of PAOL can only be established by biopsy and histopathological analysis. An adequate tissue sample is essential. An incisional biopsy is usually performed for larger lesions or those in anatomically complex locations, whereas an excisional biopsy may be feasible for small, well-defined lesions. Superficial biopsies or fine-needle aspirates are inadequate, as they may yield only necrotic or inflammatory tissue. In cases of a deep intraosseous lesion, a core needle biopsy can be considered, often augmented by flow cytometry on any aspirated material to confirm clonality and immunophenotype. Oral lymphomas are frequently initially misdiagnosed as more common conditions—underscoring the importance of having an oral pathologist or hematopathologist review the specimen. Immunohistochemical profiling is required to classify the lymphoma subtype (distinguishing DLBCL vs. PBL vs. BL vs. T-cell, etc.), usually including a broad panel of B/T-cell markers, plasma cell markers, and proliferation markers. Given overlapping features among PBL, DLBCL, BL, and even plasmacytoma, additional studies (flow cytometry or FISH for common translocations) are often needed. High-throughput molecular assays are increasingly used to identify unique biomarkers and potential therapeutic targets in these tumors [11,26,27,28,29,30,37].

### 5.4. Staging Workup

Once PAOL is diagnosed, a thorough staging evaluation is necessary to determine if the disease is truly “primary oral” (Ann Arbor stage I_E, confined to a single extranodal site) or part of more extensive lymphoma. This typically includes imaging of the entire body [38]. A PET/CT from the skull base to mid-thigh is the preferred modality for aggressive lymphomas. PET/CT is very sensitive at detecting additional hypermetabolic sites (lymph nodes, spleen, distant organs) that would upstage the disease. If PET is unavailable, a combination of contrast-enhanced CT scans of the chest, abdomen, and pelvis, plus MRI or CT of the neck, can be used. A bone marrow biopsy is advised to assess for marrow involvement, since low-volume marrow disease may not light up on PET. Routine laboratory tests are also performed (CBC, metabolic panel, LDH, and viral serologies for HIV, HBV, and HCV). LDH is often elevated in aggressive lymphoma and is a prognostic indicator. If the patient’s HIV status is unknown, testing is crucial—an unrecognized HIV infection necessitates concurrent initiation of antiretroviral therapy and prophylaxis for opportunistic infections during chemotherapy. Hepatitis B testing is also important, since rituximab-based therapy can reactivate HBV (patients would receive prophylactic antivirals). If the lymphoma is EBV-positive in a posttransplant patient, EBV DNA titers in blood might be monitored. Ultimately, if any disease beyond the oral cavity is found (nodal, visceral, or marrow), the case is not a true primary oral lymphoma but rather oral involvement of a disseminated lymphoma (stage III/IV).

### 5.5. Multidisciplinary Case Review

After staging, each case should be presented at a multidisciplinary tumor board including hematology/oncology, radiation oncology, pathology, and dental/oral surgery specialists. The pathologist confirms the subtype (and any additional molecular findings), and the hematologist uses the staging results and prognostic factors (e.g., IPI) to devise a treatment plan [39]. At this juncture, the dentist’s role shifts to optimizing the patient’s oral health prior to therapy (see next section), ensuring any potential infection sources are addressed (a process often called “dental clearance”), and planning for management of teeth within or adjacent to the tumor site (some teeth might be extracted during biopsy or later if they become non-viable).

## 6. Differential Diagnosis

The differential diagnosis for a primary oral lymphoma is broad, as many other conditions—benign and malignant—can mimic its clinical and histologic appearance. Key entities to consider are outlined in Table 5 (summarizing distinguishing clinical and pathological features):

### 6.1. Squamous Cell Carcinoma (SCC)

By far the most common oral malignancy, SCC often presents as a chronic ulcer or mass with irregular, raised borders and surface keratosis, frequently on the lateral tongue, floor of mouth, or gingiva in patients with risk factors (tobacco/alcohol) [40]. SCC can mimic lymphoma if it appears as a submucosal mass, but SCC usually shows epithelial dysplasia or carcinoma in situ at the margins and causes surface changes (rough or verrucous texture) earlier. Histologically, SCC has nests of malignant keratinizing epithelial cells (cytokeratin+, p63+). By contrast, lymphoma forms a diffuse subepithelial monomorphic lymphoid infiltrate (CD45+) without keratin pearls or dysplasia. SCC often does not respond to standard therapies for infection/inflammation; lack of response to usual treatment and a histology that does not fit carcinoma should prompt consideration of lymphoma [40]. Coexisting periodontal disease or chronic irritation can confound the picture, so a thorough biopsy and histopathological examination are needed in all cases.

### 6.2. Plasma Cell Neoplasms

An extramedullary plasmacytoma or multiple myeloma in the oral cavity can appear very similar to PBL—both may present as a soft tissue mass with plasmacytoid cells on biopsy [11,12,13,34,41]. Clues favoring a plasma cell neoplasm include older age with possible systemic signs (anemia, renal issues) or known myeloma, the presence of multiple osteolytic lesions on imaging, or detection of a monoclonal protein in serum/urine. Histologically, plasmacytoma/myeloma will show sheets of CD138^+^, CD56^+^ plasma cells that are monoclonal (restricted κ or λ light chain). By contrast, PBL tumor cells usually lack B-cell markers (CD19–, CD20–) and often are EBV+ (EBER+)—features not seen in plasmacytoma. Reactive plasma cell gingivitis, a benign condition, causes diffuse gingival enlargement with a polyclonal plasma cell infiltrate and will respond to removing the irritant or to steroids, unlike PBL [11,12,13,34,41].

### 6.3. Leukemic Infiltrates (Granulocytic Sarcoma)

Acute myeloid leukemia (especially the monocytic subtype) can present with a diffuse gingival enlargement or tumor-like mass in the oral cavity (historically called “chloroma” or granulocytic sarcoma). This is often accompanied by other signs of leukemia (e.g., cytopenias) and an abnormal blood picture. Biopsy shows sheets of myeloblasts; special stains and flow cytometry (myeloperoxidase, CD117, etc.) confirm myeloid origin [42]. Treatment requires systemic leukemia therapy, not local. Acute lymphoblastic leukemia (ALL) in children can also present with prominent gingival infiltration or oral bleeding as an early sign [43]. Even chronic lymphocytic leukemia (CLL) has, in rare cases, produced an intraoral tumor as an initial presentation [44]. Recognizing a leukemic infiltrate is critical because misdiagnosis as an isolated lymphoma can delay proper chemotherapy.

### 6.4. Indolent Lymphomas and Reactive Lymphoid Lesions

Not every lymphoid proliferation in the oral cavity is of aggressive histology. Low-grade B-cell lymphomas (like MALT lymphoma) have been reported in salivary glands or oral mucosa, often presenting as slow-growing masses or chronic swelling. They are typically CD20+ but have different histology (e.g., small mature lymphocytes forming nodules) and an indolent clinical course, so they will not be the focus of this review [45]. Reactive lymphoid hyperplasia (pseudolymphoma) can also produce a localized oral mass with a polymorphic mix of lymphocytes. These benign lesions lack clonal B-cells (polyclonal by light chain and gene rearrangement analysis) and often regress if the antigenic stimulus is removed [46]. It is crucial not to overdiagnose such reactive proliferations as aggressive lymphoma, as management is completely different.

### 6.5. Granulomatous Infections

Chronic infections (e.g., tuberculosis, deep fungal infections) can cause non-healing oral ulcers or nodules with granulomatous inflammation. These often have systemic manifestations (such as pulmonary involvement or fever) and are distinguished by identifying the causative organism with special stains or cultures (AFB for TB, GMS/PAS for fungi) [47]. No monoclonal B-cell population is present (mixed T and B cells without light chain restriction). In addition, noninfectious granulomatous diseases (e.g., orofacial sarcoidosis) can produce similar lesions; these are confirmed by biopsy showing noncaseating granulomas and appropriate lab/imaging studies [47].

### 6.6. Inflammatory and Autoimmune Conditions

Inflammatory diseases like granulomatosis with polyangiitis (GPA or Wegener’s disease) can produce destructive oral lesions that mimic lymphoma. However, they usually have other systemic features (sinonasal involvement, renal issues) and specific serologies (c-ANCA for Wegener’s). Biopsy reveals necrotizing vasculitis with granulomas, not a clonal lymphoid infiltrate [48]. No atypical T-cells expressing CD56 are present, as in NK/T lymphoma. Other vasculitic conditions (e.g., eosinophilic granulomatosis with polyangiitis) can produce oral lesions but are exceedingly rare [49]. Additionally, Langerhans cell histiocytosis (LCH) can cause lytic jaw lesions with soft tissue swelling; it is differentiated by CD1a+/S100+ neoplastic Langerhans cells and often multiple bone lesions [50].

### 6.7. Other Malignant Mimickers

Various malignancies can initially present in the oral region. Kaposi’s sarcoma (in HIV-positive patients) may appear as vascular red/purple plaques or nodules on the palate, but biopsy shows HHV-8+ spindle cells (rather than lymphoid cells) [51]. Oral malignant melanoma (rare) can present as a pigmented or amelanotic mucosal mass and is identified by melanoma markers (S100, HMB-45) [52]. Metastatic carcinomas to the jaw (from breast, lung, etc.) can manifest as jaw tumors or non-healing extraction sites; a known cancer history and cytokeratin-positive tumor cells confirm the diagnosis [53].

## 7. Treatment Strategies

Stage I_E (primary localized) oral lymphomas are treated with curative intent using systemic therapy, often combined with site-directed therapy (radiation), analogous to therapy for limited-stage nodal lymphomas. Below, we summarize current approaches for the most common PAOL subtypes—DLBCL, PBL, BL, and peripheral T-cell lymphomas (PTCL)—highlighting chemotherapy, immunotherapy, the role of transplant, novel agents, and use of radiotherapy or surgery, noting any differences in pediatric patients. Nonetheless, no trials specific to PAOL exist, leaving its applicability somewhat speculative. (See Table 6 for an overview of standard therapies.)

### 7.1. Stage I_E Diffuse Large B-Cell Lymphoma (DLBCL)

The standard treatment for localized DLBCL is combination chemoimmunotherapy with R-CHOP (rituximab + cyclophosphamide, doxorubicin, vincristine, prednisone) [54]. For limited-stage (non-bulky, low-risk) oral DLBCL, abbreviated therapy is often used—for example, 3 cycles of R-CHOP followed by involved-site radiotherapy (~30 Gy), or 4 cycles of R-CHOP alone—with excellent outcomes [55]. Higher-risk or bulky presentations usually warrant 6 cycles of R-CHOP, sometimes with radiation to bulky sites [55,56,57]. Rituximab (anti-CD20 mAb) is an integral part of therapy and has dramatically improved survival rates. Consolidative radiotherapy (≈30 Gy) is commonly added for stage I disease, especially if only 3–4 chemo cycles are given or if any residual mass remains [58,59,60,61]. Surgery has little role beyond the diagnostic biopsy (excision of the oral tumor does not eliminate the need for systemic chemo) [58,59,60,61]. Upfront autologous stem cell transplant (ASCT) is not indicated for stage I disease—first-line therapy is usually curative. Only if the lymphoma relapses would high-dose therapy with transplant be considered [58,59,60,61]. A recent advance for high risk DLBCL is the addition of polatuzumab vedotin (anti-CD79b antibody–drug conjugate) to CHOP (omitting vincristine: “R-CHP”), which in a clinical trial improved 2-year progression-free survival in frontline treatment [62]. Anti-CD19 CAR-T cells are not used first-line but are approved for relapsed/refractory DLBCL after ≥2 prior therapies. In the context of stage I, CAR-T would only be relevant if the lymphoma relapses and is chemo-refractory [63]. Recently bispecific T-cell engagers like glofitamab and epcoritamab (targeting CD20) have shown high response rates in refractory DLBCL [63,64]. These are generally salvage therapies—their role in a localized case would only arise if standard treatment failed [63,64].

### 7.2. Stage I_E Plasmablastic Lymphoma (PBL)

PBL has no established standard regimen from prospective trials due to its rarity, but CHOP-like therapy alone is considered inadequate [65]. Most experts use intensive regimens akin to those for BL or other aggressive HIV-related lymphomas [66]. Common choices are dose-adjusted EPOCH (etoposide, prednisone, vincristine, cyclophosphamide, doxorubicin, + rituximab if CD20+) or the HyperCVAD regimen (alternating high-dose methotrexate/cytarabine with CHOP-like cycles, + rituximab if CD20+) [66]. These have shown better responses than CHOP in retrospective reports. Some centers also incorporate novel agents (often in trials) targeting plasma cell pathways—for instance, bortezomib (a proteasome inhibitor) or lenalidomide, or even daratumumab (anti-CD38)—in combination with chemotherapy [65,66,67]. Consolidative radiotherapy (30–50 Gy) to the oral site is usually recommended after chemotherapy given the high risk of residual disease [65,66,67]. Due to PBL’s high relapse rate, many clinicians proceed with ASCT in first remission if the patient is fit. Allogeneic transplant is considered in relapsed or refractory cases. CAR T-cell therapy (CD19-directed) has limited applicability in PBL because the tumor cells often lack CD19; research is exploring CAR T-cells targeting alternate antigens (e.g., CD30 or CD38) [65,66,67].

### 7.3. Stage I_E Burkitt Lymphoma (BL)

BL is treated with intensive multi-agent chemotherapy regimens, similar to those used for systemic (disseminated) BL. The standard of care is a short, intensive inpatient regimen such as CODOX-M/IVAC (alternating cyclophosphamide, vincristine, doxorubicin, high-dose methotrexate with ifosfamide, etoposide, cytarabine) combined with rituximab [68,69]. Alternatively, HyperCVAD with rituximab is used by some. These regimens include CNS prophylaxis (intrathecal chemotherapy) due to BL’s high propensity for CNS spread [68,69]. Rituximab significantly improves outcomes. Consolidative radiotherapy is generally not used for BL (reserved only for any residual localized disease after chemo) [68,69]. Surgery has no curative role beyond diagnostic biopsy, since BL responds so well to chemotherapy. Upfront transplant is not part of initial therapy because first-line intensive chemo cures a high proportion of patients (even stage IV) [68,69]. If BL relapses, salvage chemo followed by ASCT can be attempted, and investigational targeted therapies or immunotherapies (like CD19 CAR T-cells) are considered. BL is highly curable with intensive therapy. Over 85% of pediatric BL patients are cured. Adults also have high cure rates (estimated >60–70% in limited-stage and ~50% even in advanced stage) with appropriate chemoimmunotherapy. Relapses, if they occur, typically happen early (within the first year) [68,69].

### 7.4. Stage I_E Peripheral T-Cell Lymphomas (PTCL)

Localized PTCL in the oral cavity is very uncommon, but reported cases include ALK+ or ALK– ALCL presenting as oral lesions, PTCL-NOS, or rarely an isolated oral NK/T-cell lymphoma of the palate. In general, treatment follows systemic PTCL protocols since even stage I PTCL carries a high relapse risk. Combination chemotherapy with an anthracycline-based regimen is standard: CHOP (cyclophosphamide, doxorubicin, vincristine, prednisone) for 6 cycles is typical [70]. Some clinicians add etoposide (CHOEP) in younger patients [71]. For CD30-positive PTCL (especially ALCL), brentuximab vedotin (BV, an anti-CD30 drug–conjugate) is added to chemo (BV + CHP regimen), significantly improving progression-free and overall survival in ALCL as shown in the ECHELON-2 trial [72,73]. After chemo, involved-site radiotherapy (30–40 Gy) to the primary site is generally recommended, because PTCL is less chemo-sensitive than B-cell lymphomas. Surgery has virtually no role aside from biopsy (complete resection is neither feasible nor sufficient due to microscopic disease). Given the high relapse rates, many experts consolidate PTCL patients in first remission with ASCT (especially for PTCL-NOS or ALK– ALCL) [72,73,74,75]. Allogeneic transplant is reserved for relapsed/refractory cases. In relapsed PTCL, targeted agents such as HDAC inhibitors (romidepsin, belinostat) or pralatrexate (antifolate) have modest activity [72,73,74,75]. CAR T-cell therapies for T-cell lymphomas are in early trials (e.g., anti-CD30 CAR T for ALCL). Localized T-cell lymphomas have intermediate outcomes. For example, stage I NK/T-cell lymphoma (nasal type) treated with combined chemoradiation has a 5-year survival of roughly 50–70% [72,73,74,75]. Stage I PTCL-NOS or ALK– ALCL have poorer outcomes (around 40–50% 5-year survival even with aggressive therapy). ALK+ ALCL responds better, with long-term survival often exceeding 70% in localized cases. Overall, T-cell lymphomas tend to have higher relapse rates than comparable B-cell lymphomas [72,73,74,75].

## 8. Prognosis

The prognosis of PAOL depends on the histologic subtype and stage, as well as patient factors (age, performance status, etc.). Being “primary oral” (stage I–E) is generally a favorable factor because disease burden is low. Many PAOL are detected at that stage because symptoms drive the patient to see a dentist relatively early. However it is useful to consider by subtype (Table 7).

### 8.1. DLBCL

With modern therapy, the 5-year overall survival for stage I_E DLBCL is on the order of 75–80%. The majority of patients with localized oral DLBCL can be cured with standard treatment [75]. The prognosis for PBL is very poor. Median overall survival is only ~1 year. Even with aggressive chemotherapy and consolidation, 2-year overall survival of ~30–40% is observed [11,12,13].

### 8.2. PBL

PBL has a poor prognosis, with median overall survival of 6–12 months. Factors contributing to worse outcomes include HIV positivity, advanced stage, and MYC translocation presence [11,12,13,66,67]. PBL is usually associated with poor prognosis and low survival, with some references citing median survival of 8 months. Prolonged remissions may occur, but PBL is a high-risk lymphoma with early relapses, chemo-refractory disease, and death from infections in immunosuppressed hosts [66,67].

### 8.3. BL

BL is highly curable with intensive chemotherapy. Over 85% of pediatric BL patients are cured. Adults also have high cure rates (≈60–70% in limited-stage and ~50% in advanced-stage) with appropriate therapy. Relapses typically occur early (within 1 year of treatment) [76].

### 8.4. T-Cell Lymphomas

Localized T-cell lymphomas have intermediate outcomes. For example, stage I NK/T-cell lymphoma (nasal type) treated with combined modality therapy yields a 5-year survival of ~50–70%. Stage I PTCL-NOS or ALK^− ALCL have 5-year survival around 40–50% even with aggressive therapy. ALK^+ ALCL has a relatively better prognosis, with long-term survival often >70% in localized disease [77].

## 9. Integrated Role of Dentists and Hematologists

Managing PAOL requires close collaboration between dental specialists and the oncology team at every stage—from diagnosis through treatment and survivorship. Dentists play a key role not only in detecting and diagnosing PAOL but also in preparing the patient’s oral health for therapy, managing oral complications during treatment, and providing long-term oral care after cancer treatment (Table 8).

### 9.1. Pre-Treatment Dental Assessment

Before cancer therapy begins, a thorough dental evaluation and necessary interventions are strongly advised. The goal is to eliminate sources of infection and optimize oral health to reduce complications during immunosuppression. This involves:

#### 9.1.1. Treatment of Infection Foci

Identify and treat any active or potential sources of oral infection. This means restoring or extracting teeth with advanced caries or chronic periapical lesions, performing deep periodontal cleanings and extracting teeth with severe periodontitis, and treating any mucosal infections (e.g., candidiasis). Impacted teeth or residual roots that could flare up during neutropenia should be evaluated for removal [78,79,80,81,82,83,84].

#### 9.1.2. Timing of Invasive Procedures

If extractions or periodontal surgery are needed, they should ideally be performed well before chemotherapy or head/neck radiotherapy starts—typically at least 2 (preferably 3–4) weeks prior—to allow sufficient healing. This reduces the risk of poor wound healing or osteoradionecrosis once the patient becomes immunosuppressed or if radiation is involved at those sites [79,80,81,82,83,84]. For patients who plan to receive head/neck irradiation, any teeth within the radiation field that are in doubtful condition should be extracted beforehand to minimize later osteoradionecrosis. If high-dose chemotherapy or stem cell transplant is planned, any elective oral surgery is best completed before treatment while the patient’s immune status is optimal [79,80,81,82,83,84].

#### 9.1.3. Preventive Measures

Perform a professional dental prophylaxis (cleaning) to reduce plaque and periodontal inflammation. Apply fluoride varnish and prescribe high-fluoride toothpaste to strengthen enamel, especially if salivary glands will be exposed to radiation. Custom fluoride gel trays can be made for the patient to use daily during radiation therapy to reduce radiation-related caries [79,80,81,82,83,84]. Ensure any ill-fitting dentures are adjusted to prevent mucosal injury during treatment. The patient should be educated on rigorous oral hygiene practices (gentle brushing, rinsing, and avoiding irritants) that they will need to follow during therapy [79,80,81,82,83,84].

#### 9.1.4. Coordination with Oncology Team

The dentist must communicate with the oncology team about dental findings and recommended interventions. If urgent dental issues (e.g., an abscess) are present, they should be addressed prior to chemotherapy. If cancer therapy must start immediately, the dental team can provide interim solutions or arrange any critical dental treatment during the brief window before severe immunosuppression develops [79,80,81,82,83,84]. The hematologist should share the chemotherapy schedule with the dentist so that any necessary dental procedures can be timed for when the patient’s blood counts will be relatively safe.

By performing this pretreatment dental optimization, studies have shown a reduction in oral complications during cancer therapy. It helps prevent treatment interruptions due to dental issues and establishes a baseline for better oral health post-treatment.

### 9.2. Oral Care During Cancer Therapy

Patients with PAOL typically receive intensive multi-agent chemotherapy (often with immunotherapy) and sometimes head/neck radiotherapy or stem cell transplants. These treatments can cause a range of acute oral side effects: mucositis (painful mucosal inflammation and ulceration), xerostomia (salivary gland damage leading to dry mouth), dysgeusia (taste changes), and infections due to neutropenia (e.g., candidiasis, herpes reactivation). Proactive supportive oral care during treatment can significantly reduce the severity of these complications and improve the patient’s quality of life.

#### 9.2.1. Preventive Oral Care Measures

Ensuring optimal oral hygiene and a clean, hydrated oral environment is essential during treatment. Basic oral care measures include gentle brushing of teeth and gums 2–3 times daily with a soft toothbrush and fluoride toothpaste, frequent bland rinsing (e.g., saline or bicarbonate solutions) to keep the mucosa clean and moist, using saliva substitutes or stimulants to combat dry mouth, maintaining adequate hydration, and adhering to a soft, non-irritating diet [85,86,87,88]. Diligent oral care can significantly reduce the severity of mucositis and other oral complications during therapy. Despite these measures, some degree of oral mucositis often occurs 1–2 weeks into chemotherapy.

#### 9.2.2. Management of Mucositis and Pain

Management of mucositis and pain is crucial. Evidence-based guidelines also support various interventions for oral mucositis, including the use of topical anesthetics, coating agents, antimicrobial rinses, and systemic analgesics to manage pain [85,86,87,88]. Patients are advised to use bland rinses (e.g., salt and baking soda solutions) frequently to cleanse the mouth and to avoid alcohol-based mouthwashes that can irritate tissues. Topical anesthetic rinses or gels (e.g., viscous lidocaine) can provide temporary relief for eating and oral care. Systemic analgesics (often opioids) may be required for severe mucositis pain. Severe cases of mucositis may benefit from advanced therapies like low-level laser therapy (photobiomodulation), which some studies show can reduce ulcer duration [85,86,87,88].

#### 9.2.3. Infection Prophylaxis and Management

If oral mucositis lesions become superinfected (e.g., by Candida or herpes simplex virus), they should be treated promptly with appropriate antifungal or antiviral medication. During neutropenic periods, the risk of oral infection is high [89,90,91,92]. The care team often prescribes prophylactic antifungals (e.g., nystatin or fluconazole) to prevent thrush and antiviral prophylaxis (e.g., acyclovir) in patients with prior herpes to prevent reactivation. The dental team should monitor for any signs of infection (such as white pseudomembranes of thrush or new ulcerative lesions) and ensure they are addressed immediately—for example, initiating systemic antifungals for candidiasis or antivirals for herpetic stomatitis [90,91,92,93,94].

Maintaining excellent oral cleanliness (as described above) is itself a key preventive strategy, as it minimizes the bioburden and potential entry points for pathogens. The dentist and oncology team must maintain close communication to promptly manage any oral issues and avoid delaying cancer treatments.

### 9.3. Post-Treatment Follow-Up and Long-Term Oral Care

After completing therapy and achieving remission, patients enter a surveillance and rehabilitation phase. During this high-risk period, the dentist should perform oral examinations approximately every 3 months (for the first two years). This allows early detection of any local recurrence (e.g., a new ulcer, mass, or persistent pain). Any suspicious finding should prompt immediate re-evaluation (imaging and biopsy) in coordination with the hematologist. After two years, follow-up intervals can be gradually extended if appropriate.

#### 9.3.1. Xerostomia and Salivary Gland Dysfunction

Even once the threat of recurrence diminishes, many PAOL survivors face chronic oral complications from their treatment. Among the most common is xerostomia (chronic dry mouth) due to salivary gland damage from radiation or chemotherapy [95]. Persistent xerostomia greatly increases the risk of dental caries and periodontal disease. Patients should use high-fluoride toothpaste or gels daily and saliva substitutes to mitigate xerostomia-related caries. The dentist may also prescribe sialogogues like pilocarpine if some salivary function remains. Long-term, meticulous oral hygiene and frequent dental checkups (every 3–4 months) are needed to prevent rampant decay [95].

#### 9.3.2. Radiation-Related Sequelae

Radiation-related late effects include trismus (jaw stiffness) and osteoradionecrosis (ORN) of the jaw [96]. Trismus results from fibrosis of masticatory muscles or TMJ ligaments in the radiation field; the dental team provides jaw stretching exercises and devices to maintain opening range. ORN is the death of irradiated bone, often precipitated by a tooth extraction or trauma within a prior radiation field. Dentists closely monitor any areas of exposed bone or non-healing extraction sites in irradiated patients [97]. Early ORN is managed conservatively with antiseptic rinses, antibiotics, and local care, while more severe cases may require surgical debridement and hyperbaric oxygen therapy (in consultation with oral surgery or ENT specialists).

#### 9.3.3. Chronic GVHD Effects

Patients who underwent allogeneic stem cell transplantation may develop chronic graft-versus-host disease (cGVHD) affecting the oral tissues [98]. Oral cGVHD often appears as lichen planus–like mucosal changes (white striations, atrophy), ulcers, and salivary dysfunction. This can cause chronic pain and sensitivity. Management of oral cGVHD typically involves topical immunosuppressants (e.g., steroid rinses or tacrolimus) to reduce inflammation, along with measures to protect the mucosa and relieve dryness and pain, in coordination with the patient’s transplant physicians.

#### 9.3.4. Secondary Malignancies or Late Complications

Although relatively rare, patients who have undergone certain chemotherapy or radiation treatments may develop secondary cancers; for instance, head and neck radiation can slightly increase the risk of developing new oral squamous cell carcinoma years later. Dentists should perform routine oral cancer screenings indefinitely for cancer survivors [99]. Additionally, if the patient received IV bisphosphonates or RANKL inhibitors due to bone involvement of lymphoma or osteopenia from therapy, they carry a risk of medication-related osteonecrosis of the jaw (MRONJ). Although lymphoma patients are generally less likely to require long-term treatment with these medications compared to myeloma patients, such treatment may still be necessary in cases involving bone metastases or the management of osteoporosis. The dental team should confirm medication history and use MRONJ prevention protocols (avoiding extractions if possible, maintaining excellent oral health to prevent the need for invasive procedures, and promptly treating any dental infections) [100,101].

#### 9.3.5. Dental Rehabilitation

Finally, dental rehabilitation is important for quality of life. Any teeth or oral structures lost due to the lymphoma or its treatment should be restored once the patient has recovered sufficiently. This may involve fabricating new dentures or obturators, relining existing prostheses, or in some cases placing dental implants (with caution in irradiated bone) to restore function and esthetics. Such rehabilitation helps the patient regain normal chewing, speech, and appearance, contributing to overall long-term well-being.

## 10. Conclusions

Primary Aggressive Oral Lymphomas (PAOL) are a challenging subset of extranodal non-Hodgkin lymphomas, with unique clinical-pathologic presentations, intricate molecular characteristics, and significant diagnostic challenges. The rarity and heterogeneity of these malignancies often result in delays in recognition, which can adversely affect therapeutic outcomes and patient survival. Effective management of PAOL requires a comprehensive, integrated multidisciplinary approach involving specialists across various fields, including dentists, oral pathologists, hematologists, radiologists, and oncologists. Dentists and oral healthcare professionals are often the first point of contact for patients with PAOL, highlighting their critical role in early detection through careful oral examinations and timely referral for advanced diagnostics. Pathologists and hematologists collaborate closely to ensure accurate histopathological classification and molecular characterization, which are indispensable for precise staging and the determination of optimal therapeutic strategies. Recent advancements in understanding the molecular genetics and tumor microenvironment of PAOL have made it possible to develop targeted and personalized therapeutic interventions. The exploration of the oral microbiome and its interaction with the local immune environment is emerging as a promising field of study, offering novel preventive strategies and enhancing existing therapeutic approaches. Future efforts should prioritize collaboration between healthcare settings and research institutions; this may enhance the translation of findings into effective therapeutics for PAOL.

## Figures and Tables

**Figure 1 cancers-17-03138-f001:**
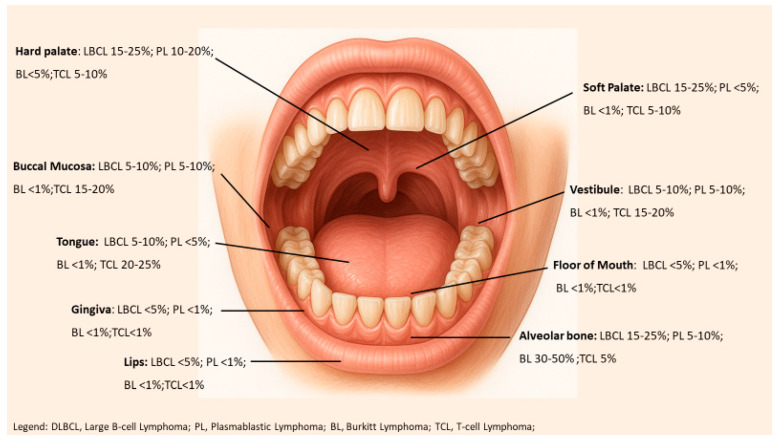
Common anatomical sites and frequency of lymphoma subtype.

**Figure 2 cancers-17-03138-f002:**
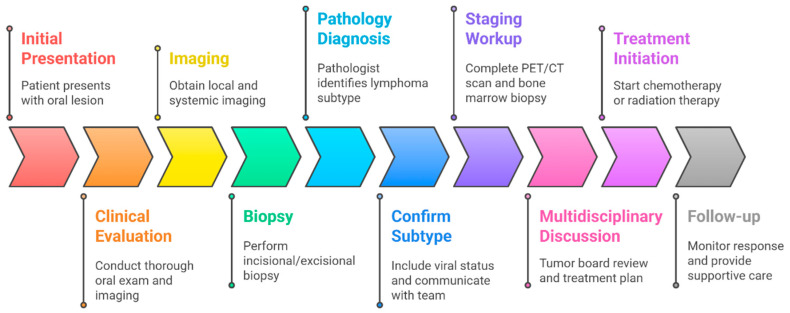
Diagnostic and treatment pathways for oral lymphomas.

**Table 1 cancers-17-03138-t001:** Anatomical distribution of main histological subtypes of PAOL. DLBCL: Diffuse large B-cell Lymphoma; PBL: Plasmablastic Lymphoma; BL: Burkitt Lymphoma; TCL: T-cell Lymphoma.

Oral Site	DLBCL	PBL	BL	PTCL
Lips	Rare. DLBCL uncommon.	Rare. Only isolated case reports	Rare.	Rare.
Tongue	Occasional. The tongue is involved in a minority of oral DLBCL cases.	Rare. Involvement by PBL is infrequent.	Rare. BL rarely presents in the mobile tongue.	Common. Tongue is one of the most frequent sites for oral PTCL, reported in 25% of cases.
Hard Palate	Common. The hard palate is a frequent site.	Common. usually present as soft-tissue masses on the hard palate.	Rare. BL seldom presents in the palate.	Occasional–Common. Palatal involvement (often destructive in nature).
Soft Palate	Frequent. Lymphomas of the tonsils/soft palate are very common for DLBCL.	Rare. PBL only rarely involves the soft palate specifically.	Rare. Primary BL of the soft palate/tonsil is very uncommon.	Occasional. PTCL can involve soft palate region, but this is not a predominant site for TCL.
Gingiva	Frequent. This is the single most common site of oral lymphoma overall.	Frequent. The gingiva is the most commonly affected site in PBL HIV-positive patients.	Common. involves the jaw and presents with gingival swelling. In non-endemic BL = 15%.	Occasional. Gingival involvement by PTCL is reported but not as predominant as for B-cell lymphomas.
Alveolar Bone	Common. Intraosseous jaw involvement is seen in many oral DLBCL cases.	Occasional. PBL can extend into or originate in jaw bones.	Frequent. Presents as destructive tumors.	Rare. Primary PTCL in the intraoral bones is very uncommon.
Buccal Mucosa	Occasional. Buccal mucosa is a less common site for oral lymphomas.	Occasional. PBL can involve the buccal mucosa, though it is not among the top sites.	Rare. BL rarely presents in the buccal soft tissue.	Common. Buccal mucosa is one of the more frequent sites for PTCL in the oral cavity.
Vestibule	Occasional. Lymphomas can present in the gingivobuccal sulcus.	Occasional. PBL is possible but uncommon.	Rare. BL in the vestibule is not typical.	Rare. PTCL involving the oral vestibule is very rare.
Floor of Mouth	Rare. Extremely uncommon.	Rare. No significant occurrences of PBL.	Rare.	Rare. PTCL is unreported.

**Table 2 cancers-17-03138-t002:** Histopathologic profiles of the major PAOL subtypes. Diffuse large B-cell Lymphoma; PBL: Plasmablastic Lymphoma; BL: Burkitt Lymphoma; TCL: T-cell Lymphoma; ENK/TL: Extranodal natural killer/T-cell Lymphoma; PTCL-NOS: peripheral T-cell Lymphoma not otherwise specified; ALK + TL: alk positive T-cell Lymphoma.

Lymphoma	Key Markers Positive	Key Markers Negative	EBV/HHV8
DLBCL	CD20, CD79a, PAX5, +/- BCL6, +/- CD10 (GCB type).	Surface Ig often present; lacks plasma markers (CD138–). May express BCL2 (variable).	EBV positive in subset (EBV^+^ DLBCL of elderly); HHV8^−^.
PBL	CD38, CD138, MUM1/IRF4, PRDM1(Blimp1), Ig light chain. Often CD79a+, CD30+. Ki-67 ~100%.	Pan-B markers CD20, CD19, PAX5 usually negative (or dim). Often CD45–. Usually CD10–, BCL6– (not GC-derived).	EBV positive in ~70%; HHV8 negative.
BL	CD20, CD10, BCL6, surface IgM, Ki-67 ~100%. MYC translocation present in all cases.	BCL2 typically negative. (CD138–, CD30–).	EBV positive in ~90% (endemic); ~30% in sporadic; HHV8^−^.
ENK/TL	CD3ε (cytoplasmic), CD56, cytotoxic granule proteins (TIA-1, Granzyme B). EBER+ in tumor cells.	Surface CD3 often^−^; usually CD5^−^, CD4^−^CD8^−^ (“double negative”). B-cell markers all ^−^.	EBV positive in >95%; HHV8^−^.
PTCL-NOS	CD3, CD5 (often partial), CD4 or CD8 (commonly CD4+). Clonal TCR gene rearrangement+.	May lose pan-T markers (e.g., CD5 or CD7^−^). No lineage-specific single marker.	EBV usually negative.
ALK + TL	CD30++, ALK-1 protein++, EMA+, often CD4+.	Usually CD3^−^ (or weak), CD5^−^, CD8^−^, CD15^−^.	EBV^−^ (typically). HHV8^−^.

**Table 3 cancers-17-03138-t003:** Genetic and Molecular Features of PAOL.

Subtype	Key Genetic Alterations	Viral Associations
DLBCL	MYC translocation in ~10%; BCL2 translocation in subset (esp. double-hit cases); BCL6 translocation in ~20%. TP53 mutations in subset (correlating with high risk).	EBV^+^ in EBV+ DLBCL, NOS (often in immunosenescence or immunosuppression); no direct HHV-8 link.
PBL	MYC translocations in ~50% (often IgH partner); Complex karyotypes common. PRDM1 (*BLIMP1*) inactivating mutations in up to 50%; other mutations: TP53 (~20–30%), JAK3, STAT3 (esp. HIV+ cases). Often exhibits MYC overexpression (even if no translocation) via NF-κB/STAT3 activation.	EBV frequently positive (~70%) (Latency I program typical). HHV-8 negative in classic PBL; but oral HHV-8+ cases occur in HIV (solid PEL variant).
BL	MYC-Ig translocation in ~100% (t(8;14) or variants). Often additional 13q or 1q abnormalities. Very few other mutations (germinal center B-cell derived): can have ID3, TCF3 mutations in sporadic BL. TP53 mutations in ~30%.	EBV^+^ in nearly all endemic BL; ~20% of sporadic BL (higher if HIV+). No HHV-8 involvement.
NK/TL	No pathognomonic translocation; commonly mutated genes: TP53, Janus kinase pathway (e.g., JAK3), *STAT5b*, *DDX3X*, *BCOR*. Also, 6q21 deletions (PRDM1, etc.) common. LOH at 17p (TP53) frequent.	EBV universal (clonal EBV in tumor cells). EBV likely causative; expresses EBV latent genes (EBNA1, LMP1 variably) and EBV miRNAs. No association with HHV-8.
ALK + TL	t(2;5)(p23;q35) *NPM1-ALK* in most; variant ALK translocations (partner genes) in others. ALK fusion protein.	Usually EBV^−^. HHV-8^−^. (Not virus-driven, unlike B-PTLD).

DLBCL: Diffuse large B-cell Lymphoma; PBL: Plasmablastic Lymphoma; BL: Burkitt Lymphoma; NK/TL:Natural Killer T-cell Lymphoma; ALK + TL: ALK positive T-cell Lymphoma.

**Table 4 cancers-17-03138-t004:** Red Flags for PAOL.

Red Flag	Description
Non-healing ulcer	Lasting > 2 weeks; indurated margins, irregular surface, not responding to topical therapy.
Red or white patches	Persistent adherent lesions that can be precancerous or malignant.
Intraoral swelling or palpable mass	Localized or infiltrative lesion without trauma or obvious cause; may include neck lymph node enlargement due to metastasis.
Unexplained tooth mobility	Tooth loosening not related to periodontal disease or trauma; often associated with underlying bone destruction.
Persistent pain or unusual sensations	Ineffective response to analgesics; may include deep pain or paresthesia (numbness, tingling in tongue, lip, or chin).
Functional difficulties	Dysphagia, speech changes, difficulty chewing or opening the mouth without clear cause.
Ear pain (referred otalgia)	Typical of tumors in the tongue or base of the mouth; absence of local ear infection.
Persistent halitosis and weight loss	Not explained by oral hygiene or diet; often a sign of advanced disease.
Persistent cervical lymph nodes	Enlarged, hard, progressive, non-tender nodes suggestive of metastasis.

**Table 5 cancers-17-03138-t005:** Differential Diagnosis of PAOL—Key Distinguishing Features.

Condition	Clinical Clues	Pathology/IHC Clues
Squamous Cell Carcinoma	Irregular ulcer or mass with raised rolled border; surface texturing (keratosis), common on lateral tongue, FOM, gingiva.	Malignant epithelial cells forming nests/pearls; IHC: Cytokeratin+, p63+. No diffuse CD45+ lymphoid infiltrate as in lymphoma. Dysplastic epithelium adjacent often present.
Plasmacytoma/ Myeloma	Often involves bone can have jaw pain; patient may have systemic signs (anemia, renal issues). Soft tissue plasmacytomas can appear as violaceous submucosal masses in older adults.	Sheets of plasma cells, eccentric nuclei. IHC: CD138+, CD56+ (common), MUM1+, monoclonal light chain in tumor; CD20–. EBER– (unlike many PBL). Serum/urine monoclonal protein usually detectable. Bone marrow involvement common in MM.
Reactive Plasma Cell Gingivitis (benign)	Diffuse gingival redness/enlargement often due to hypersensitivity. Generalized involvement rather than a focal mass; bleeds easily but patient well.	Plasma cell infiltrate polyclonal (mixed κ/λ by IHC). No atypia or destructive growth. Will lack clonal light chain restriction seen in neoplastic plasma cell lesions. Responds to removal of irritant or steroids, unlike lymphoma.
Chronic Periodontitis/ Osteomyelitis	Common; deep periodontal pockets, calculus, tooth mobility due to bone loss. Osteomyelitis can cause fistula, purulent discharge.	Periodontitis: granulation tissue with polymorphous and bone resorption; no monoclonal population. Osteomyelitis: necrotic bone with bacterial colonies, granulocytes; culture positive
Granulomatous Infection (TB, deep fungus)	Chronic non-healing ulcer or nodule. May have sinus tract (in TB of jaw), or systemic B symptoms. Regional lymph nodes often involved (scrofula). In histo/blasto, often pulmonary signs too.	Granulomas with central necrosis (TB) or organisms visible. Special stains: AFB stain+ for TB (or PCR), GMS+ for fungi. No monoclonal B-cell population; IHC CD20/CD3 will show mixed T and B without light chain restriction. Cultures confirm pathogen.
NK/T-cell Lymphoma (Nasal type)	Ulcerative lesion on midline palate often with perforation; nasal congestion/epistaxis common. Rapid progression, systemic symptoms.	Angiocentric atypical lymphoid infiltrate with necrosis. IHC: CD3ε+, CD56+, cytotoxic markers+; EBER positive (virtually 100%).
Wegener’s (Granulomatosis with polyangiitis)	Sinonasal involvement with ulceration can extend to palate; often multiple lesions, crusting, also lung/kidney involvement. c-ANCA positive in ~90%.	Necrotizing granulomas with vasculitis (fibrinoid necrosis in vessel walls). No clonal lymphocyte population; negative for EBV in lesions. PR3-ANCA blood test positive. No atypical T-cells expressing CD56 as in NK/T lymphoma.
Peripheral T-cell Lymphoma, NOS	Rare in oral cavity; may present as diffuse ulcerative stomatitis or a tumor. B symptoms present.	Diffuse infiltrate of pleomorphic T-cells. IHC: CD3+, often loss of CD5 or other B-cell markers; no EBV (unless associated with immunosuppression). Diagnosis by exclusion—requires clonality testing.
Anaplastic Large Cell Lymphoma	Very rare as primary oral lesion; might present as a solitary ulcerated nodule on the lip or gingiva. ALK+ subtype more in younger, ALK- in older.	Sheets of large anaplastic cells, pleomorphic, CD30 strongly positive. ALK IHC ± (depending on type).

**Table 6 cancers-17-03138-t006:** Therapy for Major PAOL Subtypes. DLBCL: Diffuse large B-cell Lymphoma; PBL: Plasmablastic Lymphoma; BL: Burkitt Lymphoma; TCL: T-cell Lymphoma.

Lymphoma	Chemotherapy	Immunotherapy	Radiotherapy	Stem Cell Transplant (Auto/Allo)	Targeted Agents/Novel Drugs	CAR-T Therapy
DLBCL	R-CHOP × 3–4 cycles (non-bulky) or × 6 (bulky/high IPI)	Rituximab (anti-CD20) with all cycles	ISRT 30–36 Gy (if <4 cycles of R-CHOP or residual mass)	Not indicated upfront; ASCT in relapsed refractory; Allo-SCT rarely used	Polatuzumab-vedotin + R-CHP (for high-risk first-line)	For relapsed/refractory cases (CD19+) after ≥2 lines
PBL	Dose-adjusted EPOCH or HyperCVAD; CHOP inadequate	Daratumumab or Bortezomib or brentuximab in trials	30–50 Gy ISRT after chemoCT in localized disease	ASCT in CR1 (fit patients); AlloSCT for R/R cases	Bortezomib, lenalidomide, thalidomide; PD-1 inhibitors	Limited use (CD19 often negative), only experimental
BL	CODOX-M/IVAC or HyperCVAD + rituximab; DA-EPOCH-R for older adults;	Rituximab standard (CD20+)	RT not routinely used (only in residual disease)	No role in 1st line; ASCT/AlloSCT in relapsed BL	No standard; investigational agents in R/R BL	CD19 CAR-T (salvage setting)
TCL	CHOP or CHOEP × 6; BV-CHP for CD30+ (e.g., ALCL)	Brentuximab-vedotin (CD30+) integrated into BV-CHP	ISRT 30–40 Gy to the oral site post-chemo	ASCT in CR1 (especially PTCL-NOS, ALK– ALCL); AlloSCT for relapsed or refractory cases	Romidepsin, pralatrexate, belinostat; ALK inhibitors (e.g., crizotinib in ALK + ALCL)	No approved CAR-T; CD30 or TRBC1 CAR-T under trial

**Table 7 cancers-17-03138-t007:** Summary of Prognosis for Major PAOL Subtypes. DLBCL: Diffuse large B-cell Lymphoma; PBL: Plasmablastic Lymphoma; BL: Burkitt Lymphoma; TCL: T-cell Lymphoma; NK/TL: natural killer/T-cell Lymphoma.

Lymphoma Subtype	5-Year OS	Notes
DLBCL	~75–80% (if low IPI)	R-CHOP cures majority; worse if advanced stage or double-hit (5-yr OS < 30%).
PBL	<20% (5-year); median OS ~8–12 months	Very poor despite intensive chemo; high relapse.
BL	~60–70% (5-year)	Children > 85% cured; adults > 60% if fit for intensive chemo. Rapidly fatal if not treated promptly.
NK/T-cell Lymphoma	~50–70% (5-year)	Localized disease can be cured with chemoradiation; disseminated disease <20% 2-year survival.
T-cell, ALK-positive Lymphoma	~70–80% (5-year)	Excellent prognosis in younger patients, especially with addition of brentuximab to chemo. ALK-negative ALCL ~50% 5-year.
T-cell, ALK-negative Lymphoma	~40–50% (5-year)	Generally poor with CHOP; consolidation with transplant improves some outcomes.
HIV-associated DLBCL	~50–60% (3-year)	Approaches as HIV-neg outcomes if CD4 adequate and full-dose chemo given. PBL in HIV very poor (<1 year survival).

**Table 8 cancers-17-03138-t008:** Dental interventions across all phases of PAOL treatment.

Treatment Phase	Dental Role and Interventions	Key Considerations
Pre-Treatment Dental Assessment	Thorough oral examination and radiographic evaluation (Panoramic X-ray, CBCT). Identification and management of infections, caries, periodontal disease. Extraction of non-restorable or high-risk teeth (at least 2–4 weeks prior to cancer treatment). Professional cleaning and application of fluoride varnish. Custom fluoride trays fabrication if radiation therapy planned. Oral hygiene education tailored for the upcoming oncological treatments.	Prevention of osteoradionecrosis (ORN) and infectious complications. Optimal timing is crucial to allow adequate healing and reduce risk of infection during immunosuppression.
During Oncology Treatment	Regular oral examinations to monitor for mucositis, infections, xerostomia. Management of oral mucositis (topical anesthetics, cryotherapy, photobiomodulation). Prompt diagnosis and treatment of opportunistic infections (fungal, viral, bacterial). Gentle oral hygiene practices; use of bland rinses (baking soda/saline). Coordination with oncology team regarding timing and safety of any necessary urgent dental procedures.	Minimizing oral pain and mucosal damage enhances patient comfort and nutritional status. Preventive interventions reduce complications and improve chemotherapy adherence.
Treatment Planning Considerations	Active participation in interdisciplinary tumor board meetings. Collaborative planning with oncologists, radiation therapists, and surgeons. Designing and fabricating oral protective devices (e.g., dental stents) if radiotherapy is applied. Strategic planning for preventive dental care to minimize interruptions to oncology treatments.	Collaboration reduces morbidity related to oral complications and facilitates uninterrupted cancer therapy. Protecting non-involved oral tissues from radiation and chemotherapy side-effects is essential.
Supportive and Preventive Oral Care During Cancer Therapy	Meticulous oral hygiene regimens (soft brushes, fluoride toothpaste). Prescription of saliva substitutes, saliva stimulants, high-fluoride gels. Dietary counseling (soft/bland diet recommendations). Pain management strategies including topical anesthetics, oral rinses, and analgesics. Monitoring and management of xerostomia and mucositis proactively.	Supportive oral care significantly reduces severity and incidence of oral complications. Enhanced patient adherence to oncological regimens and improved quality of life.
Post-Treatment Follow-Up and Long-Term Oral Care	Regular oral examinations every 3–4 months for the first two years post-treatment. Vigilant screening for recurrence, secondary malignancies, chronic mucosal changes. Long-term management of xerostomia (fluoride trays, saliva substitutes, dietary counseling). Management of osteoradionecrosis and trismus (mouth opening exercises, physiotherapy referrals). Psychological support through esthetic and functional dental rehabilitations (prosthetics, implants as indicated).	Early detection of recurrence improves survival outcomes. Management of chronic oral side effects significantly enhances long-term quality of life. Continuous interdisciplinary collaboration is essential for optimal patient outcomes.

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
