# Peer review of "Primary Aggressive Oral Lymphomas (PAOL): A Narrative Review of Diagnosis, Molecular Features, Therapeutic Approaches, and the Integrated Role of Dentists and Hematologists"

_cancers, 2025, doi:10.3390/cancers17193138_

Round 1
Reviewer 1 Report
Comments and Suggestions for Authors
The manuscript presents a broad and detailed narrative review on primary aggressive oral lymphomas (PAOL). The topic is relevant, the paper is generally well written, and the figures/tables improve clarity. The integration of hematological and dental perspectives is original and of clinical interest. However, the text is sometimes too expansive and would benefit from synthesis and a more critical appraisal of the available evidence.
Major Comments
Scope and methodology
Please explicitly state that this is a narrative review. This avoids confusion and justifies the descriptive and integrative approach, which is appropriate given the rarity of PAOL.
Balance of content
The histopathology, immunophenotype and molecular biology sections are very detailed and somewhat redundant. I would recommend summarizing the key morphologic and immunohistochemical and biological features of the main PAOL subtypes in a concise comparative table, rather than in a long descriptive paragraph. This would improve readability and provide readers with a practical reference (e.g., subtype, key markers positive/negative, viral association, typical clinical context). I would also summarize and harmonize the differential diagnosis section, rather than presenting a list of every pathological entity.
Therapeutic perspectives
The section on novel therapies (CAR-T, bispecific antibodies, targeted agents) needs a more critical discussion. In particular, emphasize that no PAOL-specific trials exist and that applicability remains speculative.
Conclusions and outlook
A concise final section highlighting future directions (e.g., multicenter registries, need for prospective data, translational studies) would strengthen the manuscript.
Nonetheless, I believe that the main focus of the review lies in section 9, which is disproportionately short compared to the preceding parts. This section, however, represents the truly “novel” contribution of the manuscript. I would therefore suggest reconsidering the structure of the paper, reorganizing the sections to place greater emphasis on the collaboration between hematologists and dentists, while streamlining the extensive background on disease biology and histopathology. Rather than providing long lists of features, these could be harmonized into a concise background introduction that leads to the core message of the review: the interdisciplinary role of hematologists and dentists.
Reviewer 2 Report
Comments and Suggestions for Authors
The manuscript concerns clinical and molecular aspects of primary oral lymphomas. This field was extensively studied over decades, and a lot of reviews were published till 2022. The current article is a multidisciplinary review which also involves recent data. Of special interest incidence and clinical features of OL in immunocompromised patients (e.g., following transplants). The content of this review is largely, based on the earlier reviews and describes, mainly, general facts about diagnosis and biomarkers of lymphomas, being not quite original in this respect. However, the review is well written, and should be useful for dentists, by increasing their vigilance for timely diagnosis of these rare malignancies, with broad analysis of clinical facts (Section 6, Differential diagnosis).
Remarks
The general principles of literature selection for this review are not described (may be done under Introduction). In present edition, it looks like a chapter from a handbook.
Line 215: For the reader’s convenience, the main molecular markers in Table 3 could be supplied with numbers of appropriate key references. The list of abbreviations should be checked (DLBCL is missed), and moved to the bottom of the table.
Line 221, Fig.2. The step of molecular biology studies is not seen in this flowchart. It should be between Pathology diagnosis and Confirm Subtype
In Section 5 (Diagnosis): one could pay some attention to the incidence and histological subtypes of oral malignancies in immunocompromised patients (e.g., those with HIV infection, and after organ transplants). Similarly, the authors could mention special epidemiological features of Burkitt lymphoma and other head/neck malignancies in children and adolescents.
In general, this comprehensive review is rather useful for practitioners in dentistry and oncology, and may be published after minor revisions.
Reviewer 3 Report
Comments and Suggestions for Authors
This is a well-prepared and comprehensive review that addresses an underrecognized but important clinical topic. It will be of significant value to both hematologists and dentists.
Major points.
- Expanding the discussion on prognosis would provide better balance.
- It would strengthen the manuscript to elaborate more clearly on the differences between primary oral lymphomas and systemic lymphomas. For example, in Section 4 (Molecular Pathogenesis), the discussion of MYC and BCL2 alterations seems to reflect systemic disease. Please clarify how these findings apply specifically to primary oral cases, and whether unique features are known.
- It would also be valuable to describe how clinicians determine whether a case represents a true primary oral lymphoma (Ann Arbor stage I_E) versus part of disseminated disease when additional extraoral lesions are subsequently identified.
Minor points
- Characters in Figure 1 could be more clearly shown with bigger font size.
Round 2
Reviewer 1 Report
Comments and Suggestions for Authors
Accepted